# Deciphering the Role of BCAR3 in Cancer Progression: Gene Regulation, Signal Transduction, and Therapeutic Implications

**DOI:** 10.3390/cancers16091674

**Published:** 2024-04-26

**Authors:** Dong Oh Moon

**Affiliations:** Department of Biology Education, Daegu University, 201 Daegudae-ro, Gyeongsan-si 38453, Gyeongsangbuk-do, Republic of Korea; domoon@daegu.ac.kr; Tel.: +82-53-852-6992; Fax: +82-53-852-6992

**Keywords:** BCAR3, cancer, estrogen resistance, migration

## Abstract

**Simple Summary:**

This article provides an in-depth review of the Breast Cancer Anti-Estrogen Resistance 3 (BCAR3) gene, focusing on its significant implications in cancer progression. BCAR3 plays critical roles in several biological processes that are pivotal for cancer development, including cell motility, hormone resistance, and immune system interactions. These functions make BCAR3 an important player in the pathology of cancer, affecting how cancer cells grow, spread, and resist various treatments. Specifically, BCAR3’s involvement in cell movement aids the metastatic spread of cancer, allowing cells to invade new tissues. Additionally, its role in hormone resistance is particularly crucial in the context of hormone-sensitive cancers like some breast cancers, where it helps cancer cells evade the effects of hormone therapies. The gene’s interaction with the immune system also suggests that it might influence how the body detects and fights cancer cells. Given these multifaceted roles, the review underscores the necessity for more comprehensive studies to fully understand the functions of BCAR3. It is posited that such research could unveil new therapeutic targets, potentially leading to more effective cancer treatments. The exploration of BCAR3 not only helps in understanding the complex biology of cancer but also opens up new avenues for therapeutic intervention.

**Abstract:**

This review comprehensively explores the gene BCAR3, detailing its regulation at the gene, mRNA, and protein structure levels, and delineating its multifunctional roles in cellular signaling within cancer contexts. The discussion covers BCAR3’s involvement in integrin signaling and its impact on cancer cell migration, its capability to induce anti-estrogen resistance, and its significant functions in cell cycle regulation. Further highlighted is BCAR3’s modulation of immune responses within the tumor microenvironment, a novel area of interest that holds potential for innovative cancer therapies. Looking forward, this review outlines essential future research directions focusing on transcription factor binding studies, isoform-specific expression profiling, therapeutic targeting of BCAR3, and its role in immune cell function. Each segment builds towards a holistic understanding of BCAR3′s operational mechanisms, presenting a critical evaluation of its therapeutic potential in oncology. This synthesis aims to not only extend current knowledge but also catalyze further research that could pivotally influence the development of targeted cancer treatments.

## 1. Introduction

BCAR3 is a member of the unique family of SH2-domain containing proteins, known as the NSP family, which includes three related proteins: NSP1, NSP-2/AND-34/BCAR3, and NSP-3/SHEP1/CHAT [1]. BCAR3 has become a focal point in the study of cancer biology due to its complex role in modulating cell signaling pathways that influence cancer progression and response to therapy. This review aims to dissect the multifaceted aspects of BCAR3, from its gene expression, mRNA regulation, and protein structure to its intricate functions within cells that contribute to cancer dynamics.

BCAR3’s critical functions are rooted in its expression and the post-translational structure of its protein, which affect not only the cellular architecture but also the cellular response to external signals. Understanding these foundational elements provides insights into how BCAR3 integrates into broader cellular functions and influences key processes such as cell migration, the cell cycle, and cellular responses to hormonal therapies.

At the cellular level, BCAR3 is integral to several signaling pathways, particularly those involving integrin signaling, which plays a crucial role in cancer cell migration and metastasis [2,3]. Moreover, BCAR3’s ability to mediate anti-estrogen resistance reveals its potential impact on treatment outcomes in breast cancer, highlighting its importance in the development of resistance to conventional therapies such as tamoxifen.

Furthermore, BCAR3’s involvement in regulating the cell cycle and its interactions within the immune system illustrate its broader implications in cancer therapy. These interactions not only affect tumor growth and progression but also modulate the efficacy of immunotherapeutic strategies, placing BCAR3 at a critical intersection of cancer growth regulation and immune response modulation.

Finally, this review addresses future research directions that are essential for elucidating BCAR3’s full potential as a therapeutic target. These include detailed transcription factor binding studies, isoform-specific expression profiling, and the development of therapeutic agents targeting BCAR3. Additionally, exploring BCAR3′s role within the immune context of the tumor microenvironment could unlock new therapeutic paradigms, potentially transforming cancer treatment by leveraging the body’s own immune system against tumor cells.

Through comprehensive exploration and detailed analysis, this review aims to consolidate our understanding of BCAR3, proposing new research avenues and highlighting its potential as a key target in the fight against cancer. This synthesis of BCAR3′s roles across different cellular and molecular contexts hopes to spur further investigation and innovative treatments that could significantly impact cancer therapy.

## 2. BCAR3 Gene Expression, mRNA Regulation, and Post-Translational Protein Structure

The BCAR3 gene is located on chromosome 1p22.1 and extends across over 285,000 bases (285 kb), encompassing a notable 17 exons, as recorded in the NCBI Gene database (https://www.ncbi.nlm.nih.gov/gene, accessed on 3 January 2024). This gene’s promoter region is notable for containing specific binding sites for a variety of transcription factors, including AREB6, CBF(2), CBF-A, CBF-B, CBF-C, CP1A, HNF-3beta, Ik-2, NF-Y, and STAT5A, as emphasized by QIAGEN. Although the literature does not extensively document direct research on how these transcription factors specifically influence BCAR3 expression, the identification of these sites suggests potential regulatory mechanisms. Furthermore, the role of inflammatory cytokines such as IL-1 and TNF in elevating the transcription levels of genes related to BCAR3 in certain cell lines such as thymic and fibroblast suggests a complex regulatory mechanism involving these cytokines, possibly through the activation of the mentioned transcription factors [4]. This intricate interaction beckons further investigation to decode the pathways through which IL-1 and TNF affect BCAR3 gene activity, particularly focusing on its implications in cancer biology.

The BCAR3 gene, a complex element within the human genome, gives rise to several mRNA splice variants. Foundational research initially characterized the main transcript, BCAR3α1, which encompasses the entirety of the coding sequence from exon 1 through 17 [5]. Subsequent genomic analyses, such as those from the Ensembl database, have identified three additional BCAR3 mRNA isoforms [6]. BCAR3α2 and BCAR3α3, while structurally distinct, result in proteins identical to BCAR3α1, commencing translation from exon 6. These isoforms utilize alternative starting exons for the untranslated regions, with BCAR3α2 using exon 4 and BCAR3α3 starting from exon 5. Diverging from these, the BCAR3β variant, initiated at exon 8, translates into a truncated form, lacking the 91 amino acids present at the N-terminus of the alpha isoforms. This results in the BCAR3β protein possessing a unique N-terminal configuration, although it retains the crucial SH2 domain, indicative of its retained potential in signaling interactions. The shared sequence of exons 9 through 17 across all variants implies a conserved functional core in their genomic design. Despite the truncation, the inclusion of the SH domain in all variants, including the truncated BCAR3β, starting from amino acid 154, suggests that the primary functional domain is preserved across the isoforms. The nuances of how these structural differences influence BCAR3’s cellular role, however, remain to be fully elucidated in the scientific literature.

Research on BCAR3 mRNA expression across various cancer types highlights its complex role in tumor biology. In the context of breast cancer, differential expression patterns of BCAR3 have been observed. Specifically, in triple-negative breast cancer (TNBC), higher levels of BCAR3 mRNA are associated with decreased survival rates, underscoring its contribution to tumor aggressiveness and poor prognosis [7]. Furthermore, BCAR3 expression varies among breast cancer cell lines; it is found in lower amounts in less invasive ER-positive cell lines (MCF-7 and T47D), while more aggressive, metastatic ER-negative cell lines (MDA-MB-231 and BT549) exhibit moderate to high levels. This expression pattern, coupled with the colocalization of BCAR3 and its binding partner p130Cas, suggests a pivotal role in facilitating cancer cell invasiveness [8]. Additionally, in Luminal A and B breast cancer subtypes, low BCAR3 expression, alongside BCAR1, correlates with poor prognosis, adverse lymph node status, and diminished response to hormonal therapy, indicating potential as markers for endocrine therapy resistance [9]. This discrepancy underscores the nuanced role BCAR3 plays in cancer progression and response to therapy, highlighting its dual nature. While elevated BCAR3 levels in TNBC and HNSCC are associated with increased tumor aggressiveness and poorer outcomes, its diminished expression in Luminal A and B subtypes correlates with resistance to endocrine therapy and worse prognosis. This divergence emphasizes the importance of context-specific understanding of BCAR3’s function in cancer biology, suggesting that BCAR3’s role is highly dependent on the molecular and cellular environment of the tumor.

In other cancer forms, BCAR3’s impact is equally significant but manifests differently. For instance, in head and neck squamous cell carcinoma (HNSCC), BCAR3 overexpression is linked to enhanced tumor growth, perineural invasion, and worse survival outcomes, proposing it as a valuable prognostic marker [10]. Conversely, in multiple myeloma (MM), high BCAR3 expression correlates with a favorable prognosis, including improved event-free and overall survival rates, while its low expression at diagnosis could predict early relapse, affirming BCAR3’s status as an independent prognostic factor and potential biomarker [11].

Investigations into ncRNAs have unveiled their regulatory impact on the BCAR3 mRNA. Notably, Junjie Hou and colleagues have provided significant insights into how miR-199a/b-3p curtails the spread and growth of colorectal cancer by targeting the 3′-untranslated region (3′-UTR) of BCAR3 mRNA, thereby disrupting cancer cell proliferation, migration, and invasion [12]. Similarly, research by Xiannan Meng and team has shed light on microRNA-126-5p’s ability to suppress BCAR3 expression in endometriosis [13], while Kun Zhou and his team’s discovery of a tRNA fragment, tRF5-Glu, underscores its regulatory influence in ovarian cancer by downregulating BCAR3 mRNA. This revelation not only adds depth to the molecular intricacies of ovarian cancer but also opens up promising therapeutic avenues [14].

The BCAR3 protein, with its 825 amino acid composition [7], showcases a structural complexity that includes a Src homology 2 (SH2) domain within codons 154 to 253 at the N-terminal, and a C-terminal reminiscent of guanine nucleotide exchange factors (GEFs) aligned with the CDC25 family, located between codons 699 and 812 [4,15,16,17]. SH2 domains are specialized in recognizing and attaching to peptide sequences that contain phosphotyrosine, playing a crucial role in the orchestration of signal transduction pathways [18,19]. GEFs, functioning as activators, facilitate the transition of small GTPases from their GDP-bound inactive state to the GTP-bound active state by promoting the exchange of GDP for GTP. Experimental evidence from cells overexpressing BCAR3 indicates its capacity to act as a GEF, influencing the activity of Ral, Rap1, and R-Ras [20]. The summarized illustration of BCAR3 gene regulation and protein structure is depicted in Figure 1.

## 3. The Intracellular Function of BCAR3

The BCAR1 (p130Cas) and BCAR3 interaction plays a vital role in modulating cellular signaling pathways, significantly impacting cellular processes such as adhesion, migration, and survival. This axis is crucial for the transduction of signals from the extracellular matrix into the cell, with profound implications in cancer research, notably in therapeutic resistance and tumor aggressiveness [21,22,23]. Moreover, BCAR3’s engagement with BCAR1 is instrumental in various cellular functions, enhancing p130Cas’s membrane localization, activating small GTPase [20,24], modulating Src kinase activity [25], influencing TGF-β/Smad signaling [26], and facilitating cell movement and invasion [27]. These functions underscore BCAR3’s pivotal role in cellular behavior regulation and highlight its potential as a therapeutic target in cancer treatment.

### 3.1. The Role of BCAR3 in Integrin Signaling and Cancer Cell Migration

Integrins, vital cell surface receptors for extracellular matrix (ECM) molecules, are activated by various ECM components, such as fibronectin, collagen, laminin, and vitronectin [28]. This activation recruits focal adhesion kinase (FAK) to the integrin complex through interactions with talin and paxillin, leading to FAK’s autophosphorylation at Y397 [29,30,31]. This phosphorylation site facilitates the binding of c-Src’s SH2 domain, activating c-Src [32], which remains in a closed, inactive conformation when Y530 at the C terminus is phosphorylated [33,34]. However, dephosphorylation of pY530 by protein tyrosine phosphatases (PTPs) such as PTP1B, upon recruitment to integrins, transitions c-Src to an open, active form [35].

Further, the activated FAK attracts BCAR1, which binds using its SH3 domain [36]. BCAR1 then connects with c-Src kinase through the c-Src binding (SB) domain at its carboxyl terminus, leading to phosphorylation by c-Src kinase, enhancing the BCAR1 and c-Src kinase interaction [23,36,37]. This phosphorylation not only marks BCAR1 as a substrate for c-Src but also significantly amplifies c-Src kinase activity [38,39,40].

Moreover, BCAR1’s association with BCAR3 through the SH2 domain of BCAR3, which links with sequences within the divergent helix-loop-helix (dHLH) domain of BCAR1 at the carboxyl terminus, is critical [2,4,20,41]. This interaction is essential for the enhancement of BCAR1-mediated c-Src kinase activation [41] and highlights the complex role of BCAR3 in promoting c-Src SH3 domain’s binding to BCAR1, showcasing their intricate interaction in cell adhesion and signaling [25]. The initial activation of c-Src, prompted by BCAR1 binding, is sufficient to phosphorylate BCAR3, which then activates Rap1 through GEF activity, possibly leading to further c-Src PTK activity and enhancing cell migration. Rap1, in turn, promotes cell spreading by targeting Rac guanine nucleotide exchange factors [42], which activate the WAVE complex among other actin-regulating proteins. This activation stimulates the Arp2/3 complex, leading to the rapid polymerization of the actin network and the expansion of lamellipodia. The polymerization of actin pushes the cell membrane outward, forming protrusions such as lamellipodia, thereby facilitating cell movement. Rac1 also regulates the reorganization of cell–matrix adhesions, enabling cell movement by coordinating the formation and disassembly of integrin-mediated adhesion sites. This pathway highlights the significant roles of integrin signaling, c-Src activation, and the interactions between BCAR1 and BCAR3 in cell migration, ultimately leading to the activation of Rho GTPases, pivotal in modifying the actin cytoskeleton. Furthermore, BCAR3, when monomethylated at lysine K334 (K334me1) by SMYD2, is identified by a novel methyl-binding domain in FMNL proteins, leading to the modulation of lamellipodia properties and illustrating the complex regulatory mechanisms that orchestrate cell movement [43].

It is crucial to acknowledge that the characterization of BCAR3 as a GEF has not been conclusively demonstrated. Although BCAR3 exhibits sequence similarity with the CDC25 domain characteristic of Ras family GEFs, and is implicated in promoting GTP loading for Rap1 and Rho family GTPases, there is a lack of definitive evidence to confirm BCAR3’s direct GEF activity. Notably, the activation of Rap1 might be an indirect effect, possibly mediated by Cas/C3G (RapGEF1), as suggested in the literature (PMID: 12432078). This highlights the need for a prudent interpretation of BCAR3’s role within integrin signaling complexes. The potential indirect nature of BCAR3’s role in signal transduction calls for additional investigation to elucidate these signaling pathways further.

Marie-Line Garron and her team provide groundbreaking structural insights into the BCAR3 and HEF1 (NEDD9/Cas-L) complex, revealing its significant role in anti-estrogen resistance and integrin signaling in breast cancer [44]. In the study, BCAR3 and HEF1 bind through the interaction between the C-terminal domain of HEF1 and the GEF domain of BCAR3. Specifically, HEF1 binds near the αI helix and the C-terminus of the αA helix of BCAR3’s GEF domain. This interaction is distinct from the putative GTPase binding site but aligns with regions involved in allosteric regulation, suggesting a potential role in modulating GEF domain functions. In the study by Rama Ibrahim, Antoinette Lemoine, Jacques Bertoglio, and Joël Raingeaud, it was found that overexpression of HEF1 increases colorectal carcinoma cell migration through Src-mediated phosphorylation of FAK, and mutations at specific serine residues enhance this effect due to protein stabilization [45]. They also identified BCAR3 as a critical mediator of HEF1-induced migration, with amino acid mutations that disrupt the HEF1-BCAR3 interaction impairing cell migration, suggesting the potential of HEF1 as a biomarker for tumor progression. The summarized depiction of BCAR3’s role in integrin signaling and cancer cell migration is presented in Figure 2.

In summary, BCAR3 is a critical co-regulatory network that impacts lamellipodia dynamics and supports the development of membrane ruffles, crucial for cell migration and the invasion process in cancer [2,8,27,43,46]. Overall, the extensive research highlights the therapeutic potential of targeting the BCAR3-BCAR1 and BCAR3-HEF1 interaction as a strategy for innovative cancer treatment approaches.

### 3.2. BCAR3 Triggers Anti-Estrogen Resistance

Within the field of breast cancer research, estrogen’s role is paramount, especially concerning its interaction with estrogen receptors (ERs), ERα and ERβ. These receptors, upon estrogen binding, play a pivotal role in cell proliferation and survival by initiating gene transcription critical for cell growth. This process begins when estrogen binds to one of the two primary ERs, leading to the formation of a hormone-receptor complex that translocates to the nucleus to bind estrogen response elements (EREs) to DNA. This interaction initiates the transcription of genes crucial for cell growth, division, and survival. It does so by attracting coactivators and forming a transcription complex, which leads to the production of proteins that promote cancer cell proliferation and help prevent cell death, including cyclins, growth factors, and Bcl-2 [47,48].

Tamoxifen, functioning as a selective estrogen receptor modulator (SERM), is instrumental in the treatment of estrogen receptor-positive (ER+) breast cancer. Its action mainly involves binding to ERs, blocking the proliferative actions initiated by estrogen [49,50]. Despite its effectiveness, resistance to tamoxifen represents a substantial challenge in managing ER+ breast cancer long-term. Resistance mechanisms include ER gene mutations that modify the receptor’s structure, making it less susceptible to tamoxifen’s inhibitory effects, and the activation of alternative growth signaling pathways, such as HER2/neu or PI3K/Akt [51,52]. 

Phosphorylation of ERα significantly affects its function within the cell, including its response to estrogen, localization within the nucleus, and interaction with DNA and chromatin. This phosphorylation, especially at multiple sites on ERα, plays roles in modulating the receptor’s function and its impact on drug resistance [53]. Notably, phosphorylation of serine residues at positions 102 and 104 by cyclin-dependent kinases [54,55], and serines 118 and 167 by mitogen-activated protein kinase (MAPK) and ribosomal S6 kinase (RSK), AKT, respectively [55,56], highlight the receptor’s estrogen-independent activation function [57]. Additionally, serine 236’s phosphorylation within the DNA-binding domain affects DNA interaction, and phosphorylation at serine 305 by protein kinase A (PKA) [58,59], along with tyrosine 537 in the ligand binding domain targeted by c-Src kinase, underlines the complexity of ERα regulation and its implications for tamoxifen resistance [60,61].

The BCAR3 gene, identified in 1998, has emerged as a significant factor in the context of estrogen-independent growth in breast cancer [5]. In the early stages, breast tumors largely rely on estrogen for growth and development [62,63]. This reliance provides an opportunity for anti-estrogen medications such as tamoxifen to suppress their proliferation effectively by blocking the cancer cells’ estrogen receptors, thus reducing the hormone’s capacity to stimulate tumor expansion [64,65]. However, the effectiveness of these treatments often diminishes over time as many breast cancers develop resistance to anti-estrogen interventions, presenting a complex challenge for ongoing treatment plans and highlighting the need for new therapeutic targets [66,67]. Investigations have illuminated its role in activating specific cellular pathways and proteins, such as Rac and cyclin D1, crucial in reducing the efficacy of antiestrogen therapies. Furthermore, BCAR3’s involvement in promoting epithelial–mesenchymal transition is pivotal in breast cancer metastasis and resistance. It also facilitates cell proliferation independently of estrogen by activating alternate signaling pathways that bypass the estrogen receptor pathway, establishing BCAR3 as a central figure in anti-estrogen treatment resistance and a potential marker for disease progression [68,69,70].

BCAR3’s overexpression in breast cancer cells notably boosts ERK1/2 activation through c-SRC, with phosphorylation markers indicating enhanced activity [21,71]. This, along with the BCAR3-BCAR1 interaction, activates the Phosphoinositide 3-Kinase (PI3K)/AKT pathway [24,72]. Moreover, BCAR3′s interaction with BCAR1 leads to the activation of cyclin D through Rac1 and PAK1 activation, illustrating the complex pathways through which BCAR1 and BCAR3 facilitate anti-estrogen resistance [15,68].

This interaction between BCAR1 and BCAR3 sparks the activation of key signaling molecules such as c-Src, PI3K, ERK, and cyclin D, instrumental in fostering anti-estrogen resistance. Specifically, c-Src’s role in enabling tamoxifen resistance by affecting cellular proliferation through direct phosphorylation of ERα [60,61] or activation of EGFR and STAT5b signaling [73,74,75], the hyperactivation of the PI3K and AKT pathway in cancers, ERK1/2’s phosphorylation of ERα leading to ligand-independent transcription and tamoxifen agonistic effects [76,77,78], and cyclin D1’s regulated interaction with ERα enhancing transcriptional activity all constitute critical facets in the development of tamoxifen resistance [79,80,81]. The summarized illustration of the mechanisms by which BCAR3 contributes to tamoxifen resistance is outlined in Figure 3.

In conclusion, the BCAR1 and BCAR3 interaction triggers a comprehensive signaling cascade that culminates in anti-estrogen resistance in breast cancer cells. This intricate network of signaling events underscores the complexity of cancer cell proliferation and the challenges in treating estrogen receptor-positive breast cancer. The c-Src, PI3K, ERK, and cyclin D pathways, in particular, play pivotal roles in enabling breast cancer cells to bypass the growth-inhibitory effects of tamoxifen, highlighting the necessity for novel therapeutic strategies. Understanding the molecular underpinnings of anti-estrogen resistance, especially the roles of BCAR3, opens potential avenues for therapeutic intervention. By targeting these specific pathways, there is hope for developing more effective treatments that can overcome resistance mechanisms. This could lead to improved outcomes for patients with ER+ breast cancer, offering a more tailored and effective approach to managing the disease.

### 3.3. Cell Cycle Regulation by BCAR3

Understanding the regulation of the cell cycle is crucial in the context of cancer research, as disruptions in this process are a hallmark of cancer cell proliferation. The BCAR3 gene, particularly in B-cell biology, plays a significant role in cell cycle control. BCAR3’s expression increases in response to activation signals from the B-cell receptor (BCR), indicating its importance in B-cell activation and potential implications in B-cell malignancies. BCAR3’s interaction with the scaffold proteins BCAR1 and HEF1 is essential for activating Cdc42, a GTPase that significantly impacts the regulation of the cell cycle [82]. This interaction occurs through BCAR3’s GEF domain, traditionally involved in GTPase activation, suggesting that BCAR1 and HEF1 might regulate BCAR3’s ability to interact with and activate GTPases. The complexities involved in studying scaffold protein roles highlight the challenges of experimental strategies that can lead to non-physiological outcomes [83]. The process by which BCAR3 activates Cdc42 remains an active area of research, with theories proposing that BCAR3 initially activates a Ras subfamily GTPase, which then interacts with a GEF for Cdc42. This mechanism aligns with the known regulatory crosstalk between the Ras and Rho GTPase families [84,85,86]. Cdc42 is implicated in the regulation of cell cycle-related cyclins, specifically influencing the levels of cyclin D, and potentially cyclin E and A mRNAs, through a not fully understood mechanism involving p70S6k [87]. This function of Cdc42 suggests it plays a crucial role in managing the translation of these cyclins, essential for cell cycle progression from the G1 to the S phase, which also involves Cdk4 activation, Cdk2 de-phosphorylation, and reduction of kinase inhibitors p27 and p21.

HEF1’s role in the cell cycle is multifaceted, involving interactions that ensure the correct timing of crucial cell cycle events. Its interaction with Nek2 is vital for timely centrosome separation, and deficiencies in HEF1 lead to premature activation [88]. During the G2-M phase transition, HEF1 is pivotal for activating Aurora A kinase, crucial for continuous mitotic progression. HEF1 also affects RhoA activity through its interaction with ECT2, a RhoA GEF, essential for controlling actin cortex contractility during mitosis [89,90,91]. These interactions underscore HEF1’s significant role in cell cycle regulation, particularly during mitosis, where abnormal RhoA activity can have profound implications [92]. HEF1 also interacts with zyxin and Src kinase, crucial for mitotic spindle assembly and cellular adhesion [93,94]. The degradation of HEF1 post-mitosis, leading to a decrease in its levels, invites further investigation into its controlled degradation and its timing within the cell cycle [95]. The summarized depiction of BCAR3’s role in cell cycle progression is captured in Figure 4.

In summary, the dynamic interactions of BCAR3 with BCAR1, HEF1, and various GTPases provide a complex regulatory network that is essential for normal cell cycle progression and potentially implicated in cancer when dysregulated. These findings emphasize the need for continued research into these pathways to better understand their roles in cancer biology and to explore potential therapeutic targets for B-cell related cancers.

### 3.4. BCAR3 and Immune Regulation in Cancer Therapy

The exploration of cancer vaccines utilizing peptides presented by MHC-I molecules of dendritic cell for CD8 T-cell activation has been persistently advanced, stimulated by the successes of T-cell therapies, coupled with the growing interest in personalized mutant antigens [96,97]. Vaccine peptides have traditionally been sourced from differentiation proteins, cancer-germ cell proteins, or mutated proteins. Yet, the potential of antigens derived from non-mutated proteins is often curtailed by self-tolerance mechanisms [98]. Tumor-specific mutated antigens circumvent some tolerance hurdles but introduce logistical challenges in their identification and individualized vaccine formulation, entailing considerable technical and resource demands [99,100,101]. The variability in vaccine effectiveness further complicates their application. Targeting antigens directly implicated in cellular processes such as growth, survival, or metastasis presents a strategic approach for immunotherapy, given these processes’ pivotal roles in cancer progression. Intracellular phosphorylation-modified peptides, processed and presented by MHC-I molecules on cancer cells and specifically recognized by CD8 T-cells, emerge as a novel class of neoantigens for immunotherapeutic targeting [102,103,104]. Such peptides, often originating from phosphoproteins engaged in frequently disrupted signaling pathways in cancer, signify a promising direction in cancer immunotherapy. A notable example includes the HLA-A*0201-restricted phosphopeptide from insulin receptor substrate 2 (pIRS2).

In this context, Victor H. Engelhard et al. evaluate a novel phosphopeptide, BCAR3 phosphorylated at T130 (pBCAR3126-134), probing the safety and immunogenicity of a distinctive vaccine incorporating either or both phosphorylated peptides [105]. Utilizing genetically modified mice that express a chimeric MHC molecule amalgamating human and murine components, the study examines phosphorylated BCAR3 expression in human-derived melanoma and breast cancer cell lines through Western blot analysis. Autologous mature dendritic cells, treated with these phosphopeptides, serve to activate T-cells from donor blood, demonstrating cytotoxicity towards cancer cells. In-vivo experiments reveal that dendritic cells, laden with the BCAR3 peptide, activate T-cells and suppress tumor growth in a melanoma xenograft model effectively. In the clinical phase, melanoma patients were inoculated with the BCAR3 peptide vaccine, formulated to trigger an immune response with minimal adverse effects, using an adjuvant and poly-ICLC to boost immunogenicity. The vaccine exhibited immunogenicity in both mice and human subjects, effectively curbing tumor growth in xenograft models. The clinical trial, enrolling 15 participants, reported predominantly mild to moderate adverse events, with no severe adverse events, dose-limiting toxicities, or fatalities. Notably, a 17% immunological response rate to pBCAR3126-134 was observed, underscoring the vaccine’s safety and potential in eliciting an immune response against the cancer-associated pBCAR3126-134. This encourages further exploration of phosphopeptide-directed immune therapies, with future endeavors aimed at enhancing the magnitude and durability of phosphopeptide-specific immune responses.

Macrophages are central to the immune system’s fight against cancer, utilizing the process of phagocytosis to engulf and digest cancerous cells. Once engulfed, the cancer cells are exposed to a destructive environment of enzymes and reactive oxygen species, leading to their elimination. Furthermore, macrophages release various cytokines and chemokines, attracting other immune cells to the tumor site and enhancing the immune response [106]. By presenting antigens from cancer cells to T cells, macrophages activate the adaptive immune system for a targeted attack against tumors. Additionally, through the secretion of apoptotic molecules such as TNF-alpha, macrophages exert direct cytotoxic effects on cancer cells, highlighting their multifaceted role in tumor suppression [107].

M1 macrophages are known for their tumor-fighting capabilities, activated by Th1 cytokines and microbial products [108]. They can directly destroy tumor cells through the production of reactive species and pro-inflammatory cytokines, also facilitating Th1 responses that contribute to tumor eradication. Conversely, M2 macrophages often support tumor progression through activities related to tissue repair and immune regulation, driven by Th2 cytokines. Within the tumor microenvironment, their actions can suppress immune responses against tumors, promote angiogenesis, and aid in tumor cell dissemination.

Research by Fernando O. Martinez et al. demonstrated that IL-4-induced M2 macrophages exhibit a significant increase in BCAR3 gene expression, indicating a potential link between BCAR3 and macrophage polarization towards a tumor-promoting phenotype [109]. Similarly, Chenxi Zeng et al. revealed that BCAR3 plays a role in M2 macrophage polarization via the IL-4/Stat6 pathway, suggesting a mechanism by which BCAR3 may influence the tumor microenvironment [110,111].

The interaction of BCAR3 with macrophage polarization presents a critical insight into its role within the tumor microenvironment. The upregulation of BCAR3 in M2 macrophages, which are associated with tumor progression, underscores the potential of BCAR3 as a target for therapeutic intervention. By modulating BCAR3 activity, it may be possible to influence macrophage polarization and, consequently, the immune response to tumors. Targeting BCAR3 to prevent or reverse the polarization of macrophages towards a tumor-promoting M2 phenotype could pave the way for novel cancer treatments that harness the body’s immune system. This highlights the importance of further research into BCAR3’s functions and interactions within the immune system to develop effective cancer immunotherapies. The summarized graphic representing the model of immune dynamics mediated by BCAR3 in cancer is featured in Figure 5.

Key aspects of BCAR3’s regulation of cellular functions and mechanisms related to cancer that have been discussed are summarized in Table 1.

## 4. Future Research Directions for BCAR3

### 4.1. Transcription Factor Binding Studies

Understanding the regulation of the BCAR3 gene in cancer is crucial, given its roles in modulating cell survival, proliferation, and migration. Research into how BCAR3 expression is controlled within the tumor microenvironment, where inflammatory cytokines such as IL-1 and TNF are prevalent, is particularly vital. These cytokines are known to activate various signaling pathways that can alter gene expression profiles, impacting tumor progression and therapy responses.

IL-1 and TNF, often elevated in the tumor milieu, play significant roles in inflammatory processes and have been shown to engage specific transcription factors that bind to the BCAR3 promoter region [4]. However, there is a notable gap in research regarding the specific roles these transcription factors play in regulating BCAR3 expression, emphasizing the need for more detailed studies. The promoter region of the BCAR3 gene contains binding sites for various transcription factors, including AREB6, CBF(2), CBF-A, CBF-B, CBF-C, CP1A, HNF-3beta, Ik-2, NF-Y, and STAT5A. Notably, NF-Y and STAT5A are known to be activated by inflammatory cytokines such as IL-1 and TNF, which suggests a pathway through which inflammation could regulate BCAR3 expression [115,116]. To deepen our understanding of the relationship between the activation of transcription factors NF-Y and STAT5A and the expression of the BCAR3 gene, it is essential to conduct gene silencing studies. These studies would involve systematically silencing NF-Y and STAT5A in various cancer cell lines to observe the impact on BCAR3 expression. This research approach could provide significant insights into how these transcription factors regulate BCAR3, potentially elucidating new mechanisms of cancer progression and identifying targets for therapeutic intervention.

The methodology for such a study would typically include the use of siRNA or shRNA technologies to achieve the targeted knockdown of NF-Y and STAT5A. Post-transfection, the expression levels of BCAR3 can be quantitatively assessed through techniques such as qPCR and Western blotting to measure both mRNA and protein levels, respectively. Additionally, it would be beneficial to use chromatin immunoprecipitation (ChIP) assays to directly observe whether reductions in NF-Y and STAT5A impact the binding of these transcription factors to the BCAR3 promoter region.

Furthermore, functional assays could be employed to evaluate the biological consequences of NF-Y and STAT5A knockdown on cell behavior, including changes in proliferation, migration, and invasion capabilities of cancer cells. These experiments will help clarify whether the modulation of BCAR3 expression through these transcription factors influences oncogenic properties in cellular models.

Overall, this research could significantly contribute to the field by providing a clearer picture of the molecular pathways involving NF-Y and STAT5A that regulate BCAR3. Understanding these relationships could open up new avenues for the development of targeted cancer therapies, leveraging the modulation of these transcription factors.

### 4.2. Isoform-Specific Expression Profiling

Isoform-specific expression profiling through RNA sequencing (RNA-seq) is a critical research approach for understanding the diverse roles of BCAR3 mRNA isoforms in various cancer subtypes. This method provides a comprehensive analysis of the expression levels and patterns of these isoforms, allowing for a detailed correlation with clinical outcomes and disease progression. The primary goal is to elucidate how different isoforms contribute to cancer biology and to identify potential targets for personalized therapy.

BCAR3 produces several mRNA isoforms [6]. Each isoform may have unique functions and be differently expressed in various cancer types. Understanding these differences is crucial for developing isoform-specific drugs and therapeutic strategies. By employing RNA-seq, researchers can generate a high-resolution map of BCAR3 isoform expression across a wide range of cancer cell lines and tumor samples.

The process begins with the collection of tissue samples from patients with different cancer subtypes. These samples undergo RNA extraction, followed by library preparation where RNA is converted into a library of cDNA fragments [117]. These libraries are then sequenced to generate vast amounts of data, representing the diverse RNA molecules present in the samples. Advanced bioinformatics tools are used to analyze this data, identifying and quantifying each BCAR3 isoform’s presence.

Subsequently, statistical analysis helps correlate the presence and levels of specific BCAR3 isoforms with various clinical parameters, such as patient survival rates, tumor grade, response to treatment, and recurrence rates. This analysis might reveal, for instance, that one isoform is predominantly expressed in aggressive tumor types and associated with poorer outcomes, suggesting a potential role in tumor progression.

Moreover, isoform-specific expression profiling can aid in understanding the molecular mechanisms of BCAR3 action in cancer cells. It can reveal whether certain isoforms are involved in activating or inhibiting specific signaling pathways known to influence cancer cell behavior. For example, differences in the signaling pathways activated by each isoform could explain variations in cancer cell motility and invasiveness, offering insights into how these isoforms contribute to metastasis.

Finally, the integration of RNA-seq data with other molecular data, such as protein expression and mutation analysis, provides a more complete picture of cancer biology. This integrated approach can identify biomarkers for disease and help predict which patients are most likely to benefit from specific treatments, moving toward a more personalized medicine approach.

### 4.3. Therapeutic Targeting

In the realm of oncology, the development of inhibitors targeting the BCAR3 gene presents a promising therapeutic avenue, particularly given BCAR3’s implicated role in cancer cell migration, survival, and proliferation. However, to date, no specific inhibitors directly targeting BCAR3 have been developed. This gap underscores the critical need for novel therapeutic strategies that can modulate BCAR3 activity in cancerous cells.

One potential strategy for developing BCAR3 inhibitors is through Structure-based Virtual Screening (SBVS). This method utilizes three-dimensional structures of the BCAR3 protein to computationally screen and identify potential compounds that can bind and inhibit its activity. SBVS combines the precision of molecular docking processes with the breadth of virtual compound libraries, offering a cost-effective and efficient approach to identify new leads without the need for extensive physical screening [118,119].

Once potential inhibitors are identified through SBVS, their therapeutic effects must be rigorously validated in preclinical models. This involves testing the candidate compounds in various cancer cell lines and animal models to evaluate their efficacy in inhibiting BCAR3-mediated pathways. Essential assessments include measuring the impact on cell proliferation, survival, migration, and metastasis. Additionally, it is crucial to evaluate the side effects and toxicity of these compounds to ensure they are safe for further development.

The effectiveness of these inhibitors can be quantified through various in-vitro and in-vivo assays. In-vitro methods include cell viability assays, Western blotting for pathway activation, and migration assays, while in-vivo tests might involve using xenograft models where human cancer cells are implanted in immunocompromised mice to study the compound’s effect on tumor growth and metastasis spread.

Overall, the development of BCAR3 inhibitors is still at a nascent stage, and extensive research is needed to translate these early-stage discoveries into viable cancer therapies. The use of SBVS and thorough preclinical testing are integral steps in this process, potentially leading to breakthroughs in cancer treatment strategies.

### 4.4. Exploring the Role of BCAR3 in Immune Cell Function within the Tumor Microenvironment

The role of BCAR3 in immune cells related to cancer is a critical area of research with potential implications for therapeutic strategies. Immune cells such as T cells, B cells, macrophages, dendritic cells, and natural killer cells play pivotal roles in the immune response to cancer [120,121]. These cells can either suppress or promote tumor growth, and their function is influenced by their ability to navigate the complex signaling environment of the tumor microenvironment. Despite its known roles in cell adhesion and migration in cancer cells, the function of BCAR3 in immune cells remains poorly understood, highlighting a significant gap in the current research landscape. There is a compelling need to investigate the expression and functionality of BCAR3 in immune cells within the tumor microenvironment. This could provide new insights into how tumors manipulate immune responses and potentially reveal new targets for immunotherapy. The expression of BCAR3 might influence key immune processes such as the migration of immune cells into the tumor, their survival within the immunosuppressive tumor milieu, and their ability to execute an effective anti-tumor response.

To thoroughly investigate BCAR3 in immune contexts, several research methods are recommended. Gene expression analysis through quantitative PCR or RNA sequencing could determine the levels of BCAR3 in different immune cells from cancer patients. Protein-level investigations using Western blot or flow cytometry could assess BCAR3 activation and its downstream signaling effects. Functional assays, including cell migration and cytotoxic assays, would elucidate the role of BCAR3 in the effector functions of immune cells. Additionally, immunohistochemical staining in tumor sections could visualize BCAR3 expression relative to immune cell markers, providing insights into its spatial and functional relevance in the tumor microenvironment.

By deepening the understanding of BCAR3’s role in immune cells, researchers can unlock new avenues for cancer therapy that manipulate BCAR3-mediated pathways to boost the immune attack on tumors or counteract the immune evasion strategies employed by cancer cells. This line of inquiry not only fills a crucial research void but also holds promise for enhancing the efficacy of immunotherapies in oncology.

### 4.5. Identifying Roles in Cancer Stem Cells

The interplay between BCAR3 and cancer stem cells (CSCs) remains an underexplored terrain, particularly in the domain of breast cancer where CSCs exhibit unique characteristics. These cells are hypothesized to be the root of cancer initiation, capable of self-renewal, and are often implicated in resistance to conventional therapies and metastatic progression [122,123]. Notably, the phenotype of breast cancer CSCs, characterized by the expression markers CD44+/CD24−/low and their ability to form mammospheres in culture, sets them apart as targets for therapeutic intervention [124,125].

Despite the recognition of BCAR3’s role in cancer cell signaling and its potential in modulating therapy responses, research has yet to delve into its functions within the context of CSCs. The absence of such studies creates a critical knowledge gap, as elucidating BCAR3’s role could unveil novel strategies to target these resilient cell populations.

Investigating BCAR3 in CSCs is pivotal to comprehend how it might influence the self-renewal and differentiation processes inherent to these cells. BCAR3’s known functions in signal transduction suggest it could play a role in the maintenance of the CSC phenotype or their ability to resist therapies. Its potential involvement in pathways commonly associated with stemness and survival, such as Wnt, Notch, and Hedgehog, could also be of significant interest.

To address this, a research methodology would include isolating CSCs from breast cancer cell lines and patient-derived tumors, followed by the characterization of BCAR3 expression and function. Advanced molecular techniques such as CRISPR/Cas9 gene editing could be employed to manipulate BCAR3 levels and decipher its impact on the hallmarks of CSCs. Functional assays to assess changes in drug resistance, tumorigenicity, and metastatic capacity in response to BCAR3 modulation would offer insights into the gene’s role in these processes.

The anticipated outcomes of such research are multifold. They may reveal BCAR3 as a modulator of CSC properties, providing a new lens through which we can understand cancer resistance and recurrence. Furthermore, if BCAR3 is implicated in the maintenance of CSCs, it could emerge as a promising target for therapies aimed at eradicating this challenging cell population, potentially leading to more durable responses in breast cancer treatment.

This research promises not only to fill a significant void in our understanding of BCAR3’s role in breast cancer but also sets the stage for the development of novel CSC-targeted therapies. Given the pivotal function of CSCs in cancer pathogenesis, such advancements could revolutionize treatment paradigms, aligning with the shift towards precision medicine in oncology.

## 5. Conclusions

In this review, we have explored the multifaceted roles of BCAR3 in cancer biology, emphasizing its regulation at the genetic, transcriptomic, and protein levels and its involvement in critical cellular processes. The findings underscore the gene’s intricate functions in signaling pathways that are pivotal for cancer cell migration, growth, and survival, particularly through mechanisms such as integrin signaling and modulation of cell cycle dynamics.

BCAR3’s role in anti-estrogen resistance particularly highlights its potential as a therapeutic target in hormone-responsive cancers, such as breast cancer. By influencing estrogen receptor pathways, BCAR3 not only contributes to the complexity of treatment responses but also suggests a pathway that could be exploited for therapeutic gains. Moreover, the gene’s involvement in cell cycle regulation and its interactions with immune cells within the tumor microenvironment have opened new avenues for research and potential treatment strategies that could leverage the immune system against cancer.

Emerging studies pinpointing BCAR3 hyperexpression in aggressive neoplastic forms suggest novel prospects for biomarker-centric therapies. Pharmacotherapeutics targeting hyperactivated BCAR3-centric signaling networks, particularly through its liaison with integrins and SH2-domain proteins, are conceivable. Concurrently, expanding insights into BCAR3’s synergism with proteins such as HEF1 augments the possibility of engineering polypharmacological agents capable of dismantling these elaborate signaling edifices. Elucidating BCAR3’s comprehensive therapeutic promise necessitates expansive preclinical and clinical investigations. Such endeavors should prioritize the discovery of efficacious BCAR3 inhibitors, decode its complex signaling interactions, and ascertain the neoplastic subtypes wherein BCAR3 antagonism would be optimally efficacious. Moreover, examining the synergy of BCAR3 inhibitors within established therapeutic regimens warrants exhaustive inquiry. Furthermore, BCAR3’s involvement in oncogenic stem cells and the burgeoning immunotherapy arena is critically impactful. Targeting these progenitor cell populations, the suspected nexus for relapse and metastatic dissemination, may markedly enhance prognoses. Exploiting BCAR3’s immunoregulatory capacity could intensify the therapeutic impact of cancer immunization and cellular therapies.

Future research directions outlined in this review, such as transcription factor binding studies, isoform-specific expression profiling, and the development of BCAR3-targeted therapies, are crucial for deepening our understanding of BCAR3’s roles and exploiting them for clinical benefit. The investigation into BCAR3’s impact on immune cell function within tumors could particularly transform therapeutic approaches by integrating immunotherapy with targeted molecular interventions.

Conclusively, BCAR3 emerges not only as a critical molecular player in cancer pathways but also as a promising target for innovative treatments. Its diverse roles across different aspects of cancer biology highlight the importance of continued comprehensive research to harness this potential fully. This endeavor will likely lead to more effective, personalized cancer therapies, significantly impacting patient care and outcomes in oncology.

## Figures and Tables

**Figure 1 cancers-16-01674-f001:**
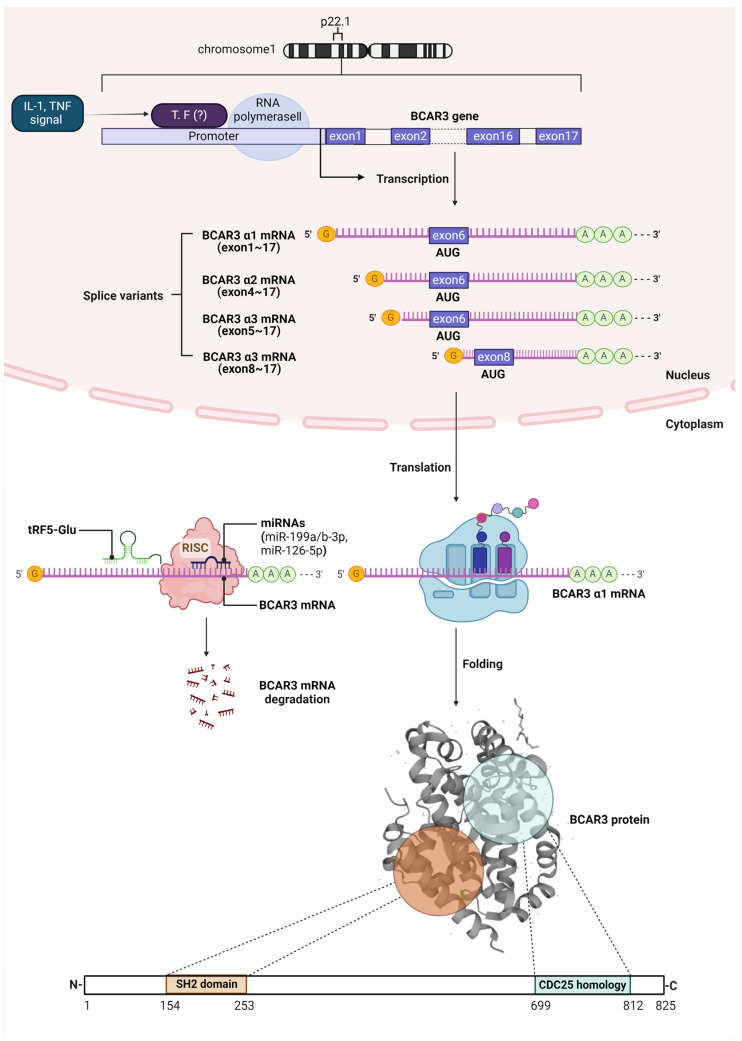
BCAR3 gene regulation and protein structure. The schematic provides a detailed overview of the BCAR3 gene structure and its transcriptional regulation, post-transcriptional mRNA control, and the resulting protein’s interaction within breast cancer cells. Located on chromosome 1 at p22.1, the BCAR3 gene consists of 17 exons and is responsive to inflammatory signals, which can induce transcription through various transcription factors. Post-transcriptional regulation is depicted by the influence of microRNAs, such as miR-199a/b-3p and miR-126-5p, and tRNA fragments such as tRF5-Glu, which target the mRNA for degradation, modulating the levels of BCAR3 protein synthesis. Upon translation, the BCAR3 protein forms a multifaceted structure endowed with an SH2 domain at its N-terminus, spanning amino acids 154 to 253, and a CDC25 homology domain resembling guanine nucleotide exchange factors at the C-terminus, between residues 699 and 812. These domains facilitate BCAR3’s interaction with BCAR1, which contributes to several key cellular activities, including the modulation of small GTPase activation, regulation of Src kinase activity, and involvement in the TGF-β/Smad signaling pathway. These interactions are crucial for breast cancer cells to navigate the complexities of cellular signaling, aiding in cell movement, invasion, and, particularly, the development of resistance to anti-estrogen therapies such as tamoxifen. Cartoon in Figure 1 was created with https://BioRender.com and accessed on 2 February 2024.

**Figure 2 cancers-16-01674-f002:**
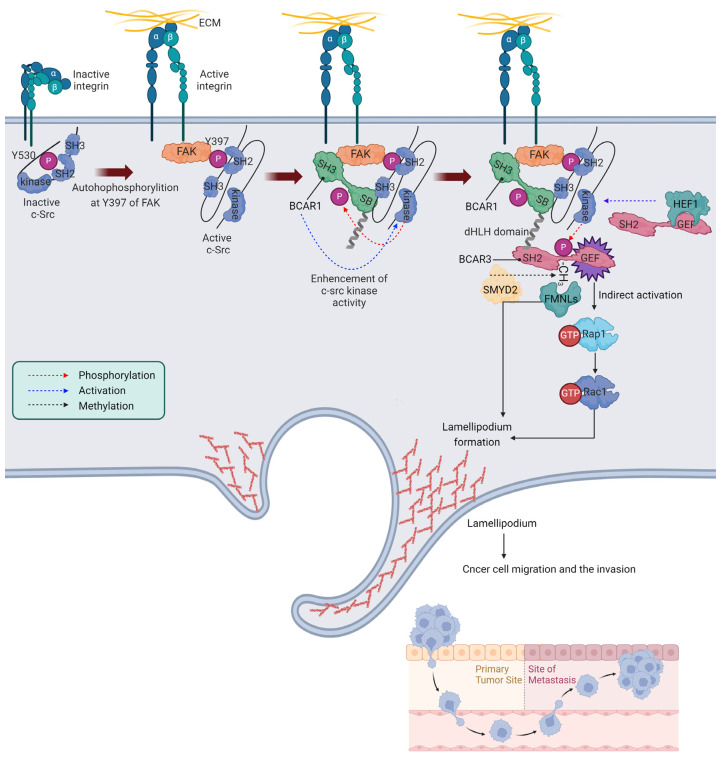
BCAR3 in integrin signaling and cancer cell migration. Activated integrins recruit focal adhesion kinase (FAK), which is autophosphorylated at Y397. This event facilitates the binding and activation of c-Src via its SH2 domain. The active c-Src interacts with BCAR1, phosphorylating it to enhance the interaction between these two proteins. BCAR1 then binds to BCAR3, which is linked through its SH2 domain to the dHLH domain of BCAR1, pivotal for c-Src kinase activation enhancement. This complex formation is essential for lamellipodia dynamics and the formation of membrane ruffles, processes fundamental to cell migration and invasion in breast cancer. Activation of c-Src by BCAR1 binding is sufficient to phosphorylate BCAR3, leading to Rap1 activation and subsequent Rac1 stimulation, facilitating cell movement. BCAR3, when monomethylated at K334, is recognized by FMNL proteins, which are recruited to the cell periphery by SMYD2-mediated methylation to modulate the properties of lamellipodia. The interaction of BCAR3 and HEF1 increases Src-mediated phosphorylation of FAK on Tyr-861 and 925. Cartoon in Figure 2 was created with https://BioRender.com and accessed on 3 February 2024.

**Figure 3 cancers-16-01674-f003:**
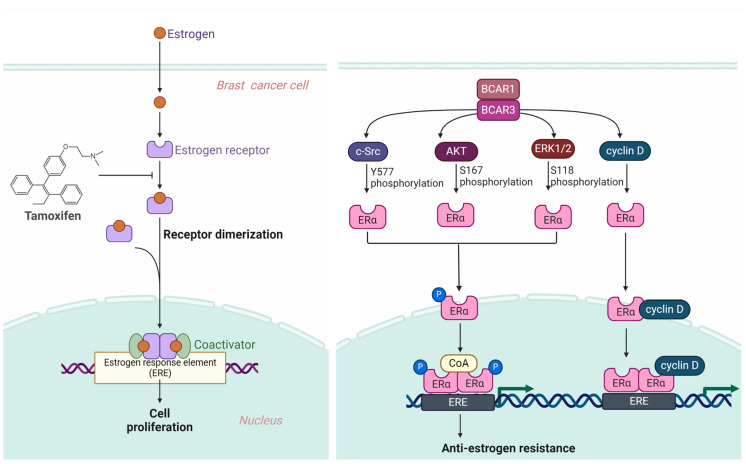
Mechanisms of tamoxifen resistance: The role of BCAR1 and BCAR3 in breast cancer cell signaling. This diagram delineates the complex interplay between estrogen-mediated signaling and the induction of anti-estrogen resistance by BCAR1 and BCAR3 pathways in breast cancer. Under normal circumstances, estrogen binds to the estrogen receptor alpha (ERα), facilitating receptor dimerization and subsequent interaction with coactivators to initiate transcription of genes that drive cell proliferation. Tamoxifen interferes with this pathway by inhibiting ERα, thereby hindering the gene transcription necessary for cell proliferation. Despite the presence of tamoxifen, the proteins BCAR1 and BCAR3 circumvent this blockade by activating alternative signaling cascades involving c-Src, AKT, and ERK1/2, as well as modulating cyclin D levels. This results in the phosphorylation of ERα, which can then proceed to bind estrogen response elements (EREs) and promote gene transcription that leads to cell proliferation. The schematic underscores the resilience of breast cancer cells in the face of anti-estrogen therapies and highlights the critical roles of BCAR3 in fostering this resistance. Cartoon in Figure 3 was created with https://BioRender.com and accessed on 4 February 2024.

**Figure 4 cancers-16-01674-f004:**
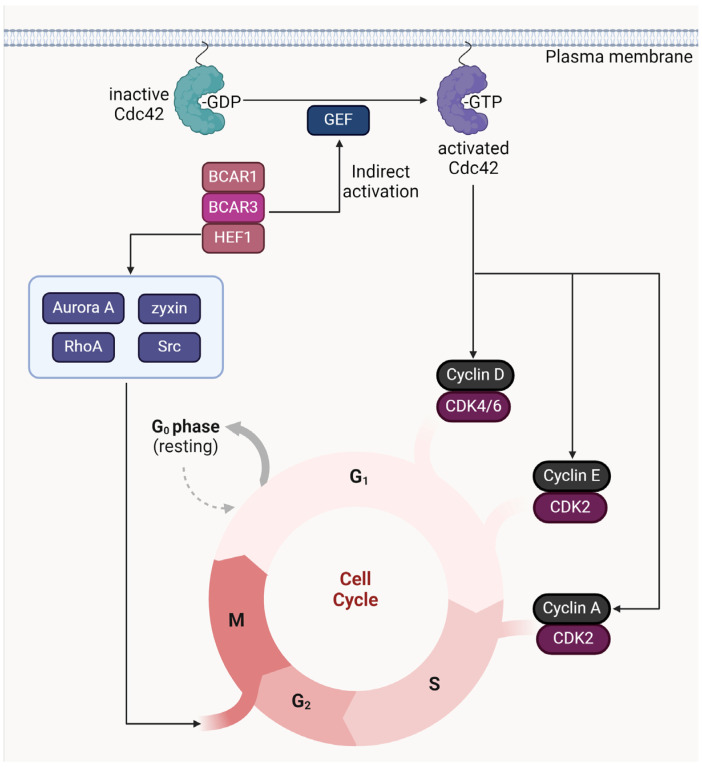
Model of BCAR3 in cell cycle progression. Inactive Cdc42 bound to GDP is activated by the guanine nucleotide exchange factor (GEF) domain of BCAR3, a process facilitated by the scaffold proteins BCAR1 and HEF1. The activation of Cdc42 by BCAR3 plays a crucial role in regulating cell cycle transition from the G1 to S phase by influencing the levels of cell cycle-related cyclins D, E, and A. This regulation is mediated via a complex network, potentially involving p70S6k. Additionally, the diagram highlights the interactions between HEF1 and other cell cycle regulators such as Aurora A, RhoA, and Src, which contribute to the correct timing of cell cycle events such as centrosome separation and mitotic progression. The figure underscores the significance of BCAR3’s multifaceted role in the cell cycle, particularly in the context of cancer research, where dysregulation of these pathways can have profound implications. Cartoon in Figure 4 was created with https://BioRender.com and accessed on 5 February 2024.

**Figure 5 cancers-16-01674-f005:**
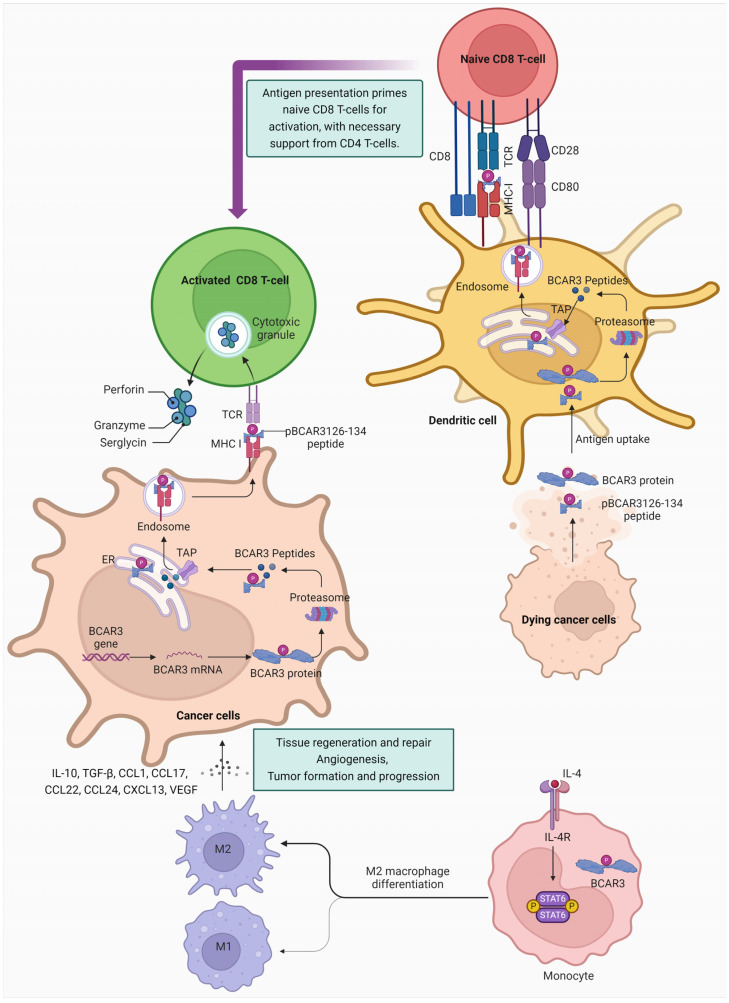
Model of immune dynamics mediated by BCAR3 in cancer. The depicted cascade showcases the immune mechanisms initiated by BCAR3 peptides in cancer. It outlines the initial engagement of dendritic cells, which internalize antigens and present the phosphorylated BCAR3126-134 peptide on MHC I complexes. Naive CD8 T-cells, upon interaction with MHC I-peptide complexes and receiving co-stimulatory signals from CD4 T-cells, undergo activation. These primed CD8 T-cells then unleash cytotoxic molecules such as perforin, granzyme, and serglycin to induce apoptosis in cancer cells. Concurrently, cancer cells transcribe the BCAR3 gene into mRNA, translating into BCAR3 proteins that are processed into peptides by the proteasome. These peptides are then presented on the cancer cell surface via MHC I molecules, marking the cells for immune surveillance and destruction. The figure also demonstrates the dichotomy of macrophage polarization: pro-inflammatory M1 macrophages versus anti-inflammatory M2 macrophages, induced by different cytokines, such as IL-4 in the case of M2, which fosters tissue repair but may also facilitate tumor growth and metastasis through the release of cytokines, chemokines, and angiogenic factors. This interplay between cellular and molecular entities underscores the complexity of the tumor microenvironment and the potential of targeting BCAR3 in cancer therapy. Cartoon in Figure 5 was created with https://BioRender.com and accessed on 6 February 2024.

**Table 1 cancers-16-01674-t001:** Comprehensive Analysis of BCAR3’s Cellular Function Regulation and Cancer-Related Mechanisms.

Model (Cell/Animal)	Methods	Cellular Function of BCAR3	Ref.
Esophageal cancer cells and xenograft mouse model	Loss-of-function assays. Xenograft growth and lung metastasis studies. Bioinformatics analysis.	CircBCAR3 promotes esophageal cancer cell proliferation, migration, and invasion and inhibits ferroptosis in vitro. In vivo, it supports the growth and metastasis of esophageal xenografts.	[112]
Ovarian cancer cells	Deep sequencing. RNA binding assays. Proliferation assays. Use of tRF5-Glu mimics.	tRF5-Glu regulates BCAR3 expression by binding to its 3′UTR, leading to decreased BCAR3 levels. Lower BCAR3 expression is associated with suppressed ovarian cancer cell proliferation.	[14]
Breast cancer cell lines (MCF-12A cells)	Microinjection of anti-BCAR3 antibody. siRNAs targeting BCAR3 and SH2 domain of BCAR3.	BCAR3, particularly through its SH2 domain, plays a crucial role in EGF-induced DNA synthesis, indicating its involvement in cell cycle progression and mitogenic signaling pathways.	[113]
Head and neck squamous cell carcinoma (HNSCC) patients	RNA-sequencing and bioinformatics analysis. siRNA transfection.	BCAR3 is overexpressed in HNSCC and contributes to tumor growth. Its expression is associated with perineural invasion (PNI) and poor survival.	[10]
Breast cancer cell lines (estrogen-independent 578-T, estrogen-dependent MCF7, ZR-75-1)	Overexpression studies. F-actin analysis. Kinase activity assays. Luciferase assay for cyclin D1 promoter activation.	BCAR3 overexpression leads to activation of Rho family GTPases Cdc42 and Rac, resulting in changes in F-actin distribution, enhanced PAK1 autophosphorylation and kinase activity, and activation of the cyclin D1 promoter.	[15]
TNBC cells, mouse orthotopic tumor models	In-vivo and in-vitro studies. Analysis of RNA expression databases. Correlation analysis. Functional assays.	BCAR3 is significantly upregulated in TNBC and contributes to tumor growth and progression. Elevated BCAR3 levels are associated with poor patient survival. BCAR3, in conjunction with MET receptor signaling, regulates proliferation and migration of TNBC cells.	[7]
Mammary-epithelium specific SMYD2 ablation in mice, breast cancer cells, PDX (Patient-Derived Xenografts), genetically engineered mice	In-vivo survival studies. Identification of physiological substrates. Methylation signaling pathway analysis. In-vitro migration and invasiveness assays.	BCAR3 is identified as a physiological substrate of SMYD2, and its methylation at K334 is crucial for metastasis. Methylation of BCAR3 by SMYD2 recruits FMNL proteins to cell edges, influencing lamellipodia dynamics and promoting cancer cell migration and invasiveness.	[43]
Breast cancer cell lines (MCF7 cells)	Gain-and loss-of-function approaches. Adhesion signaling and spreading assays.	BCAR3 modulates c-Src activity and regulates the association between Src and BCAR1. This coordination is crucial for breast cancer cell adhesion signaling and spreading, contributing to aggressive and invasive tumor phenotypes.	[16]
Breast cancer cell lines (MCF7 cells)	Use of structure-based BCAR1 and BCAR3 mutants. ERK1/2 activity assays. Reverse-phase protein array.	BCAR3’s antiestrogen resistance critically depends on binding to BCAR1. The BCAR1/BCAR3 interaction enhances BCAR1 phosphorylation, potentiating antiestrogen resistance. This resistance is associated with increased ERK1/2 activity.	[21]
Breast cancer cells (MCF-7, MDA-MB-231, BT-549 and SK-BR-3 cells)	Coimmunoprecipitation. Automated imaging system for cell migration.	BCAR3 inhibits the TGFβ/Smad signaling pathway, leading to suppression of Smad activation, gene transcription, cell migration, and matrix digestion.	[26]
Human glomerular mesangial cells	Adenovirus-mediated gene transfer. Co-precipitation assays.	BCAR3 associates with CrkII in response to endothelin-1, a process that is enhanced by Pyk2 activity. This interaction is implicated in proliferative kidney pathologies and is part of the ET-1 signaling pathway, suggesting a role in cell proliferation, contraction, and extracellular matrix synthesis in the kidney.	[114]
PTPα-null MEF cells	Reconstitution assays. Localization studies.	BCAR3 acts as a connector between phospho-Tyr789 PTPα and BCAR1 at adhesion sites, facilitating BCAR1 interaction and phosphorylation by Src. This identifies a novel role of BCAR3 in promoting cell migration through the assembly and activation of integrin-induced adhesions.	[3]
Normal epithelial cells	Localization studies. Phosphorylation assays. Lamellipodia dynamics assessment. Cell migration assays.	BCAR3 is essential for regulating cytoskeletal dynamics during cell migration, necessary for Cas phosphorylation and lamellipodia dynamics, crucial for cell migration, and part of a positive-feedback loop with Cas for cellular localization and activation signaling.	[2]
Melanoma and Breast Cancer Cell Lines, Mice, Human Donors, Clinical Trial Participants	Clinical trial with vaccinations using pBCAR3126-134 and pIRS21097-1105 peptides in adjuvant and Hiltonol. Adverse events monitoring based on NCI CTCAE. Interferon-γ ELISpot assay for T-cell responses.	pBCAR3 peptides triggered immunogenicity in vivo and in vitro. T-cells specific for pBCAR3126-134 inhibited tumor xenograft growth., Identification of pIRS21097-1105 peptide in human tumors by mass spectrometry. Induction of T-cell responses in clinical trial participants with no severe AEs, DLTs, or deaths. Immune response to pIRS21097-1105 in 42% of patients and to pBCAR3126-134 in 17% of patients.	[105]
1878 MM patients (1930 samples) from 7 independent datasets	Comparative analysis of BCAR3 expression in different stages and molecular subtypes. Analysis of 1q21 amplification. Assessment of event-free survival (EFS) and overall survival (OS). Therapeutic response evaluation to bortezomib and dexamethasone.	Predictor of better prognosis in MM patients, associated with higher EFS and OS. Indicative of prognosis post-relapse. Independent prognostic factor. Potential biomarker.	[11]
32 cases of ectopic and eutopic endometrium in endometriosis, 31 controls	Real-time PCR. Immunohistochemistry. Western blotting. Lentivirus overexpression. Vector knockdown. CCK-8 assay. Transwell experiments. Estrogen intervention experiments.	Promotes migration and invasion of endometrial cells in endometriosis. Associated with higher expression in advanced stages of endometriosis. Does not induce EMT directly. Regulates anti-estrogen effects.	[13]

## Data Availability

The data presented in this study are available on request from the corresponding author due to (specify the reason for the restriction).

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
