# Peer review of "Deciphering the Role of BCAR3 in Cancer Progression: Gene Regulation, Signal Transduction, and Therapeutic Implications"

_cancers, 2024, doi:10.3390/cancers16091674_

Round 1

Reviewer 1 Report

Comments and Suggestions for Authors

This review summarized literature on a not-so well researched signaling molecule that deserved further attention. The style is very readable and touches on all areas of pertinent research. Style is a little verbose. Figures are clear and detailed. Future directions section touches on multiple directions but could be more focused.

-              While the potential for multiple splice variants is described, how much of the N-terminus of BCAR3 is lost in the shorter, beta, version (does it impact on SH domain) and what impact might splicing have on translation of the alternate full-length alpha versions?

-              A concern is the early work addressing the role oof BCAR3 as a guanine nucleotide exchange factor (GEF). Although it harbors sequence similarity with CDC25 domain of Ras family GEFs, and promoted the loading of Rap1 and some Rho family GTPases with GTP, there are no definitive demonstrations of direct BCAR3 GEF activity. For the relate CHAT/NSP3 it was reported that Rap activation was likely secondary to Cas/C3G (RapGEF1) PMID: 12432078. Rac activation is also discussed nicely in introduction of PMID: 17427198. Some text to acknowledge this uncertainty and modification of figures2 and 4 to reflect potential indirect activation of Rap1 and Cdc42, respectively, would be helpful to avoid misleading.

-              Reference 43 has been published in a peer-reviewed journal

Author Response

This review summarized literature on a not-so well researched signaling molecule that deserved further attention. The style is very readable and touches on all areas of pertinent research. Style is a little verbose. Figures are clear and detailed. Future directions section touches on multiple directions but could be more focused.

-              While the potential for multiple splice variants is described, how much of the N-terminus of BCAR3 is lost in the shorter, beta, version (does it impact on SH domain) and what impact might splicing have on translation of the alternate full-length alpha versions?

⟶ Thank you for your insightful query regarding the splice variants of BCAR3, particularly the impact of splicing on the N-terminus in the shorter beta variant and its potential effects on the SH2 domain, as well as the implications for the translation of the full-length alpha variants. Figure 1 has been updated to include a depiction of the mRNA splice variants of BCAR3, and Section 1 now clarifies the inclusion of the SH2 domain within the beta variant.

-              A concern is the early work addressing the role oof BCAR3 as a guanine nucleotide exchange factor (GEF). Although it harbors sequence similarity with CDC25 domain of Ras family GEFs, and promoted the loading of Rap1 and some Rho family GTPases with GTP, there are no definitive demonstrations of direct BCAR3 GEF activity. For the relate CHAT/NSP3 it was reported that Rap activation was likely secondary to Cas/C3G (RapGEF1) PMID: 12432078. Rac activation is also discussed nicely in introduction of PMID: 17427198. Some text to acknowledge this uncertainty and modification of figures2 and 4 to reflect potential indirect activation of Rap1 and Cdc42, respectively, would be helpful to avoid misleading.

⟶ Thank you for the constructive feedback on the role of BCAR3 as a guanine nucleotide exchange factor (GEF). In response to the concerns raised, Section 3.1 of the manuscript has been revised to incorporate a detailed discussion of the uncertainties surrounding BCAR3’s GEF-like activity. This revision acknowledges the current literature suggesting that BCAR3 may facilitate GTP loading on Rap1 and Rho family GTPases indirectly, rather than exhibiting direct GEF function.

Additionally, Figures 2 and 4 have been updated to more accurately represent the potential indirect activation mechanisms of Rap1 and Cdc42 by BCAR3, respectively. These changes aim to present a more accurate representation of BCAR3's role in integrin signaling and ensure that the visual aids within the manuscript do not lead to misconceptions.

-              Reference 43 has been published in a peer-reviewed journal

⟶ Thank you for the update on Reference 43.

Reviewer 2 Report

Comments and Suggestions for Authors

Manuscript entitled "Deciphering the Role of BCAR3 in Cancer Progression: Gene Regulation, Signal Transduction, and Therapeutic Implications"

Major issues:

1. The authors should provide a table summarizing the impacts of BCAR3 in various cancers, regarding the biology and clinical evidence.

2. The authors should have more discussion on the therapeutic value of BCAR3.

Comments on the Quality of English Language

Acceptable

Author Response

Major issues:

  1. The authors should provide a table summarizing the impacts of BCAR3 in various cancers, regarding the biology and clinical evidence.

⟶ Your request has been added to Table 1.

  1. The authors should have more discussion on the therapeutic value of BCAR3.

⟶ Your request has been incorporated into the discussion section. Thank you for your valuable feedback, which has significantly contributed to enhancing the depth and quality of the discussion.